# Association between Dietary Habits, Food Attitudes, and Food Security Status of US Adults since March 2020: A Cross-Sectional Online Study

**DOI:** 10.3390/nu14214636

**Published:** 2022-11-03

**Authors:** Aljazi Bin Zarah, Sydney T Schneider, Jeanette Mary Andrade

**Affiliations:** 1Food Science and Human Nutrition Department, University of Florida, Gainesville, FL 32611, USA; 2Community Health Sciences Department, College of Applied Medical Sciences, King Saud University, P.O. Box 10219, Riyadh 11433, Saudi Arabia

**Keywords:** adults, dietary habits, food attitudes, food security status

## Abstract

Since COVID-19, global reports indicate changes in dietary habits and food security status of the population. As a follow-up to an earlier study conducted in 2020, the purpose of this online cross-sectional study was to examine food security and food attitudes and their subsequent impact on dietary habits since March 2020 and potential associations with demographics, health characteristics and lifestyle habits on dietary habits. Participants (*n* = 2036) responded to a 71-item online survey conducted between February–March 2022. Frequency counts and percentages were tabulated, and multivariate linear regressions were conducted to examine associations. Results showed that most participants indicated no change in dietary habits (45.9–88.8%) for the listed food and beverage items. A significant positive association for food attitudes scores (1.11, 95% CI 0.93 to 1.29; *p* < 0.001) and food security scores (0.53, 95% CI 0.35 to 0.71; *p* < 0.001) on total dietary habits was found. Further, significant positive associations were seen with various health characteristics such as medical conditions (*p* = 0.01) and lifestyle habits such as preparing meals at home (*p* < 0.001). A negative association was observed with females on total dietary habits (*p* < 0.001). It is necessary to encourage adults in the US to maintain the positive dietary and lifestyle habits they acquired since March 2020 in their daily living. Future studies should investigate the impact these habits have on their health long-term and sustained positive dietary and lifestyle habits.

## 1. Introduction

Coronavirus, also known as COVID-19, has contributed to over 1 million deaths and over 96 million infected in the United States (US), with the highest mortality rates associated with those who had respiratory issues (48.6%), hypertension (18.3%,) or diabetes (15.0%) [1]. This virus has led to hardships, as in early March 2020, business sales dropped, and food prices increased, causing a projected rise in food insecurity [2]. For instance, for a week during 15 March 2020, it was recorded that 3.3 million people lost their jobs, with an estimated 26 million jobs lost in the first five weeks of the confinement period [3]. The effects of confinement on the labor market were severe and impacted many adults in the US in the past two years since March 2020; nonetheless, the unemployment rate has decreased from 5.8% in 2021 to 3.6% in 2022 [4].

Along with the financial issues caused by COVID-19, issues pertaining to the population’s access to food also developed. During the rise of the pandemic, many consumers took part in panic shopping, thus over-purchased foods that led to supply chain issues and possibly the elevation in food prices [5]. The types of foods commonly purchased during panic shopping tended to be packaged, processed, and overall, less expensive, thus providing more fats, sugars, and sodium [6,7]. For example, one cross-sectional observational study demonstrated that 43.8% of participants (*n* = 3313) reported an increase in consumption of sweets such as candy, cake, cookies, and pie [8]. This is similar to other studies that showed that consuming foods considered high in saturated fats, added sugars, and sodium increased during this time [9,10,11,12,13,14]. On the other hand, due to safety concerns, consumers may have been cooking and eating more at home and/or participating in meal kit delivery services [15]. These safety concerns may have been the primary reason for an increase in online grocery shopping since COVID-19, as was reported in a meta-analysis that included a total sample of 20,538 participants [16].

Beyond challenges associated with accessing and obtaining food, mental health issues increased due to adults’ isolation and loneliness throughout COVID-19. A cross-sectional observational study (*n* = 20,215) discovered that a majority (77.7%) of the participants stayed at home 50% to 95% of the time since March 2020 [17]. The authors reported that staying at home for long periods of time, which may limit social interactions, can play a major part in growing mental health issues. Individuals that were most prone to mental health challenges due to COVID-19 included elders, academics, healthcare workers, children and teenagers, and people with a family psychiatric history [18]. Furthermore, lifestyle habits like sleeping, exercise, and eating may have been altered during COVID-19 [19]. Based on observational studies, most respondents decreased dining in restaurants and grocery shopping in the store; and increased levels of ordering takeout or delivery meals from restaurants and using electronic devices [8,20,21,22,23]. The authors had conducted a prior study in early 2020; thus, as a follow-up study, the current aim was to examine food security and food attitudes and their subsequent impact on dietary habits and potential associations with demographics, health characteristics and lifestyle habits on dietary habits since March 2020. 

## 2. Materials and Methods

### 2.1. Study Design and Participants

This cross-sectional study was conducted online through QualtricsXM, an online survey platform from February–May 2022. Recruitment occurred through ResearchMatch [24] and was voluntary and anonymous. Adults were eligible to participate if they were above the age of 18, could read in the English language, and had lived in the US since March 2020. Adults who did not meet these inclusion criteria were excluded from this study. All participants provided their informed consent prior to completing the survey. A total of 2053 adults initially participated, and 2036 completed the entirety of the survey (see Figure 1). All study protocols were granted ethical approval by the University of Florida Institutional Review Board # 202001147.

### 2.2. Survey

Participants responded to a 71-item survey that was used in a previous COVID-19 study (Appendix A) [8]. Briefly, this survey included questions about demographics (8 items), health information (10 items), lifestyle habits (11 items), dietary habits (30 items), food attitudes (6 items), and food security status (6). The researchers (A.B.Z. and J.M.A.) developed this questionnaire using adapted validated instruments [25,26,27]. The total length of time to complete the survey was 10 min.

For the survey, demographic items included age, sex, race/ethnicity, education level, employment status, geographic location of residence, and current time spent at home. Health information items were self-reported by the participants. They included weight changes, health conditions, supplement use, and if participants had followed a diet since March 2020, current height reported in feet and inches and weight reported in pounds for the researchers to calculate Body Mass Index (BMI) (body mass (kg)/height (m^2^)) and interpreted based on criteria of the Centers for Disease Control and Prevention [28].

#### 2.2.1. Lifestyle Habits

Participants indicated if certain lifestyle habits such as physical and social activities increased, decreased, did not change, or never participated prior to or during March 2020. Compared to the survey conducted during confinement in 2020 by the researchers, additional questions were added to this part of the survey, including using meal-kit services, ordering take-out or delivery services, preparing meals inside the home, and grocery shopping in stores compared to online due to input from a pilot study of adults (*n* = 25). The reliability of this revised instrument was Cronbach α = 0.65 compared to the previous instrument at Cronbach α = 0.52, which is considered acceptable [29]. Furthermore, the Pearson correlation coefficient value for each item was >0.25, which indicates acceptable validity [29]. The scoring of this instrument was the same as the prior study [8], in which a response of no change or never participated received a score of 0. For the added item of ordering take-out or delivery services plus the items from the previous study that were considered unfavorable to health, a score of 1 was provided [11,30]. If participants indicated a decrease in those activities from that additional item and the prior ones or an increase in the previous activities plus the additional items of using meal-kit services, preparing meals more in the home and grocery shopping in stores, it resulted in a score of 2, as these activities were considered favorable to health. Total scores ranged from 0–24, with higher scores resulting in favorable lifestyle habits.

#### 2.2.2. Dietary Habits

Participants responded to a total of 30 items about if their dietary habits increased, decreased, did not change, or never consumed since March 2020, which was based on the Dana-Farber’s Cancer Institute Eating Habits Questionnaire [25]. As the original instrument contained 61 items and had different responses (e.g., daily, 1–3 times weekly), the modification was made to minimize potential survey exhaustion from participants [31,32]. The reliability of this revised instrument was Cronbach α = 0.89, which is considered acceptable [29]. Furthermore, the Pearson correlation coefficient value for each item was >0.25, which indicates acceptable validity [29]. As was completed in a previous study [8], participants who indicated they did not change or never consumed that food/beverage received a score of 0, and participants who responded an increased consumption of food/beverage items that were considered nutrient-dense obtained a score of 2 for each item or a 1 if they decreased or increased consumption of foods/beverages considered energy-dense [33,34]. The total scores ranged from 0–60 points, with higher scores indicating more nutrient-dense foods/beverages consumed.

#### 2.2.3. Food Attitudes

Participants also responded to 8 statements regarding food attitudes since March 2020 from increase, decrease, or no change (never had these thoughts) based on the modified Yale Food Addiction Scale [26]. The reliability of this revised instrument was Cronbach α = 0.88, which is considered acceptable [29]. Furthermore, the Pearson correlation coefficient value for each item was >0.25, which indicates acceptable validity [29]. Total scores ranged from 0–16, in which participants who had no change or never had these thoughts obtained a score of 0, an increased response received a score of 2, and a decreased response received a score of 1.

#### 2.2.4. Food Security

Household food security was measured using the validated 6-item shortened USDA Food Security Module [27,35]. This instrument was modified to include ‘since March 2020′ prior to each item with reliability of Cronbach α = 0.78, which is considered acceptable [29]. Pearson correlations of >0.25 for each item indicated acceptable validity [29]. The instrument was scored on a scale from 0 to 6, with lower scores indicating high or marginal food security and higher scores representing food insecurity [27].

### 2.3. Statistical Analysis

Frequency counts and percentages were tabulated for demographic variables and dietary habits. Multivariate linear regression was conducted to examine associations between food insecurity and attitudes (confounders) with dietary habits (interest variable) (Table 1). An additional regression was conducted that focused on the confounding variables–demographics, health characteristics, and lifestyle habits on dietary habits. Statistical significance was determined at *p* < 0.05. All statistical analyses were conducted using JMP SAS v16 (JMP^®^, Version 16. SAS Institute Inc., Cary, NC, USA, 1989–2021) [36].

## 3. Results

### 3.1. Study Population

The sample consisted of 2036 respondents, although not all participants were required to respond to all demographic or health characteristics questions. Out of those who responded to these statements, the majority were white (83.3%), female (77.1%), held a bachelor’s degree (34.4%) and were employed full-time (45.5%). Most respondents were between the ages of 30 to 49 years old (33.5%) and married (48.2%). A slight majority of the participants lived in the South Atlantic region (21.4%) and lived with one other person (39.7%). An almost equal number of participants reported that they have stayed in their homes 50% to 75% of the time (38.8%) and 75% to 95% of the time (38.9%) since March 2020 (Table 2).

#### Health Characteristics and Anthropometrics

For participants who responded to these questions, health characteristics and anthropometrics revealed that based on the calculated body mass index (BMI) kg/m^2^, many participants were considered overweight/obese (57.6%). Most participants indicated that their weight increased since March 2020 (44.6%), did not try a diet (62.9%), nor take any supplements (68.8%). Of those reporting that they took supplements, the majority were taking four or more (42.7%). Respondents also reported having two (29.2%) or three or more medical conditions (25.1%) (Table 3).

### 3.2. Dietary Habits

The average score for total dietary habits was 15.74 ± 10.88, with a range of scores from 0 to 51. No change in dietary habits for the food and beverage items included in the survey was reported by most participants (45.9–88.8%). Although, for certain foods and beverages, participants reported increased consumption of water (42.0%); sweets, including cakes, cookies, and pies (38.5%); coffee or tea (35.3%); nuts or seeds (33.9%); non-starchy vegetables (33.2%); fruit (31.8%); potato chips or salty snacks (31.5%); peanut butter or other nut butter (25.1%); white rice or pasta (24.6%); and fish and shellfish (20.8%). For other food and beverage items in the survey, participants reported a decreased consumption of beef, pork, or lamb (27.6%); processed meats such as bacon, hot dogs, luncheon meats, sausage (24.7%); French fried potatoes (19.9%); and white bread including pita bread (15.8%) (Table 4).

### 3.3. Association between Food Security Status and Food Attitudes on Dietary Habits

On average, the total food attitudes score was 2.47 ± 2.81, ranging from scores of 0 to 12. The average score for food security was 1.17 ± 2.62, with a minimum score of 0 and a maximum score of 10. Multivariate linear regression was used to determine the significant positive association between food attitudes score with total dietary habits score (1.11, 95% CI 0.93 to 1.29; *p* < 0.001) along with a positive association between food security score and total dietary habits score (0.53, 95% CI 0.35 to 0.71; *p* < 0.001). Furthermore, significant positive association with medical conditions (0.13, 95% CI 0.03 to 0.22; *p* = 0.01), tried a diet (1.53, 9% CI 0.56 to 2.49; *p* < 0.001), and nutritional supplement intake (2.55, 95% CI 1.57 to 3.54; *p* < 0.001) with total dietary habits. A significant negative association was discovered between the female sex and total dietary habits score (−1.97, 95% CI −2.98 to −0.95; *p* < 0.001) (Table 5).

Additional multivariable associations were studied with total dietary habits scores. A positive association was found between physical activity (1.24, 95% CI 0.61 to 1.87; *p* < 0.001), preparing/cooking meals in the home (0.99, 95% CI 0.41 to 1.57; *p* < 0.001), meal kit services (1.18, 95% CI 0.48 to 1.88; *p* < 0.001), take-out/delivery of meals from restaurants (1.06, 95% CI 0.33 to 1.78; *p* < 0.001), grocery shopping in the store (1.14, 95% CI 0.52 to 1.76; *p* < 0.001), reading/studying (0.81, 95% CI 0.31 to 1.30; *p* < 0.001), sleeping hours and quality (1.09, 95% CI 0.50 to 1.67; *p* < 0.001), smoking (2.13, 95% CI 1.07 to 3.20; *p* < 0.001), and using electronic devices (2.04, 95% CI 1.01 to 3.08; *p* < 0.001) with total dietary habits score (Table 6).

## 4. Discussion

This study sought to examine food security and food attitudes and their subsequent impact on dietary habits since March 2020 and potential associations of demographics, health characteristics, and lifestyle habits with dietary habits. Results from this study showed that the association between dietary habits with food attitudes and food security continued to impact US adults in various ways. Even though a majority of participants indicated their dietary habits did not change, for those who did indicate a change, they increased their consumption of water, coffee or tea, salty and sweet foods/snacks, and decreased their consumption of white bread, red and processed meats. Furthermore, multiple lifestyle changes such as physical activity, grocery shopping in the store, meal kit services, and preparing/cooking meals at home positively impacted dietary habit scores. Factors such as medical conditions, tried a diet, and nutritional supplements intake had a significant positive relationship with dietary habit scores. In contrast, females negatively affected dietary habit scores.

Regarding dietary habits, females may have differed more than males, possibly due to preferences, and social and environmental factors [37,38,39]. As demonstrated in COVID-19 specific studies related to dietary habits, Hassen and colleagues demonstrated that females (*n* = 511) and males (*n* = 484) had different consumption patterns dependent on the country of residence. For example, in Morocco, females consumed more food due to fear, stress, and anxiety over COVID-19 compared to males, but in Egypt, males consumed more comfort food than females [40]. Another study that focused on 3 European countries–Denmark, Germany, and Slovenia, of a total of 2680 adults, showed that there were differences in consumption habits of males and females of foods/beverages that increased or decreased. For example, in Denmark, females increased their consumption of fresh fruits and vegetables, cakes and biscuits, and sweets and chocolate, whereas males decreased their consumption of fresh meats, fresh fish, cakes and biscuits, and increased consumption of canned foods [41]. However, the results of this study must proceed with caution as females were overrepresented as slightly more than 77% participated. Moreover, this study found that most adults who reported a change in dietary habits decreased their consumption of red and processed meats such as bacon and hotdogs, white bread, and French-fried potatoes. Along with a decrease in these items, there was a reported increase in consumption of alcohol, low-carbonated beverages, eggs, chicken or turkey, potatoes, starchy vegetables, and salty snacks. In addition, a greater daily intake of water, coffee or tea, immune-enhancing beverages, nuts, fruits, non-starchy vegetables, oils, and sweets such as cake, cookies, and pie were observed. In contrast to the survey conducted during the confinement in 2020 within the US population, there was an increase in the consumption of immune-enhancing beverages while a decrease in white bread and French-fried potatoes [8]. Immune-enhancing beverages may have increased due to participants’ perceptions of reducing the risk for illness. The greater intake of sweets remained significant during and post-March 2020, and alcohol consumption remained high in both surveys, possibly due to anxiety from the pandemic or returning to work [42,43]. Findings from this study were consistent with other studies, Kyprianidou et al. [44] and Caso et al. [45], regarding the increased consumption of alcohol, nuts, and oils. Moreover, Kyprianidou et al. [44] explored changes in dietary and lifestyle habits through two observational studies during (*n* = 1460) and after (*n* = 1043) the COVID-19 confinement. According to their findings, post-confinement consumption of fruit/vegetables was 11% less than during confinement. This contradicts findings from this study that participants have continued increasing fruits and vegetables consumption (31.8% and 33.2%), respectively, since March 2020. Compared to the results of this study, Alvarez-Gómez et al. [46] revealed that Spanish consumers (*n* = 510) increased their intake of red meat post-COVID19 confinement due to the increased access to local markets within the country, which conflicts with the findings from this study. Overall, though, Alvarez-Gómez et al. [46] discovered that healthy dietary and lifestyle habits, including an increase in fruit and vegetable consumption by 27% and 21%, respectively, occurred compared to pre-pandemic consumption. The higher intake of fruit and vegetables could be explained by accessibility, availability in the market, and continuing dietary habits acquired during the confinement period. Even though findings from various studies demonstrated changes in dietary habits since March 2020, very few have been conducted compared to when the confinement period occurred.

Lifestyle factors contributed positively to participants’ dietary habits. For instance, although 42.3% of participants indicated that their physical activity decreased since March 2020, physical activity was found to have a significant positive relationship with dietary habits (*p* < 0.001). As many participants in this study were not at home as frequently since March 2020, it may have affected their physical activity habits due to less time available. Moreover, if someone was active since the confinement period, potentially their dietary habits may have largely remained unchanged in which they were already consuming nutritious foods. Kyprianidou et al. [44] reported that in the Cyprus population, physical activity had decreased after the COVID-19 confinement, which aligns with this current study. Additionally, results from this study revealed that medical conditions contributed positively to dietary habits; this could be related to the healthier dietary and lifestyle habits developed during the confinement due to the participant’s awareness towards nutritious habits may lower their risk for other health problems, complications, and death. Although, no data was collected on these markers to confirm health status since March 2020. Furthermore, positive dietary habits may have been caused by more cooking at home, which causes an increase in fruit and vegetable consumption and less socializing outside the home [47,48].

Indeed, preparing/cooking meals in the home were significantly increased, as shown in this study (*p* < 0.001). This could be due to participants decreasing socialization outside the home and enjoying cooking habits acquired since March 2020. Moreover, meal kit services were significantly associated with positive dietary habits (*p* < 0.001), which could explain the increase in cooking meals at home. Meal-kit services are usually based on a subscription service, where the individual may customize meals to fit dietary preferences and receive pre-packaged fresh ingredients throughout the week. Based on a study conducted in Australia that compared 5 meal-kit delivery services with a total of 60 recipes discovered that the recipes were adequate in providing both macro and micronutrients. However, some modifications can be made to these meal-kits such as reducing sodium required in the recipes, providing more fiber-based foods and reducing the fat content, specifically saturated fat [49]. As in this study, it is unknown the frequency or types of meals ordered from these meal-kit services, relating the increased intake of certain foods from the dietary habits to the increased ordering of these services, cannot be made. The findings from this study are consistent with those made by Alvarez-Gómez et al. [46], who indicated that most people did not order food at home after confinement, showing that cooking at home was a practice that increased in frequency since confinement in Spain. Furthermore, Filimonau et al. [50] revealed the same results in England households where most enjoyed cooking at home during and after national confinement and agreed to eat out less post- confinement.

This study showed low overall food security and food attitude scores, indicating that participants were food secure and had positive attitudes towards food. The average total food security score was 1.17, and the average total food attitude score was 2.50. According to the USDA’s economic research service, food consumption trends in 2021 likely reflect the improved household income resulting from the economic recovery [51]. All food prices at stores were higher in July 2022 than in July 2021 [51]. Even though prices went up the most for eggs (38%) and fats and oils (20%), and poultry (16%), this study revealed that adults were consuming more of these foods, which could be related to the fact that the study sample consisted primarily of full-time working, highly educated, single individuals who were less affected by food prices than other demographic categories. Lastly, food attitude and dietary habit scores were positively correlated, which may be attributed to the demographics of the study participants as opposed to the entire US population. Lower food attitude scores indicated positive dietary habits. In fact, the effects of COVID-19 and social and economic restrictions are complicated and involve many factors that can cause different groups of people to act in different ways [52,53].

Although life in the US is slowly reverting to pre-COVID, 71% of respondents indicated that the socialization they participated in outside the home had decreased. Most participants in this study did not shop for groceries online; thus, grocery shopping in the store was significant on total dietary habit scores (*p* < 0.001), which may be due to preference, the atmosphere, and experience of shopping in physical stores [54]. The ability to personally select food is the primary driver of the in-store benefit [54]. Additionally, when it comes to grocery shopping, food quality has emerged as the most crucial factor to consider [54]. These results were slightly different from a study conducted in mainland China, in which results revealed that despite an increase in online grocery shopping, purchasing food in person at local supermarkets or small shops remained the most common way to obtain food in the post-lockdown period [55].

Some drawbacks of the present study must be acknowledged. First, due to the survey being conducted online, those who did not have access to a computer or the internet could not participate; therefore, selection bias resulted from the sample size and was not representative of a broader population [31,56]. This may have skewed the results with lower socioeconomic groups potentially consuming different dietary habits and participating in other lifestyle habits than those in higher socioeconomic groups. Therefore, the results from this study could not be generalized to the entire US population. Second, as a result of participants not being required to answer every survey item and may have provided inaccurate self-reporting information such as height and weight, this may have led to inaccuracies in the calculation of BMIs. Furthermore, the self-reporting bias could be from participants not remembering accurate information, wanting to look like they were in better economic standings than they were, or attempting to appear healthier based on social desirability bias [31,56].

## 5. Conclusions

This was the first study in the U.S. to examine changes in dietary habits, food attitudes, and security status since March 2020. The results of the present study indicated an increase in the consumption of fruits, vegetables, nuts, and immune-enhancing beverages, as well as an increase in home-cooked meals, and a decrease in the consumption of red and processed meat, and a limit on dining out and socializing outside the home since March 2020. Furthermore, as of March 2020, a significant percentage of the population had healthier lifestyle habits, such as engaging in preparing meals at home. It is crucial to encourage adults in the US to maintain the positive dietary and lifestyle habits they acquired since March 2020 in their daily living, which could have long-term health benefits. Future studies should investigate the impact these habits acquired since March 2020 have on their health long-term and if these positive dietary and lifestyle habits are sustained.

## Figures and Tables

**Figure 1 nutrients-14-04636-f001:**
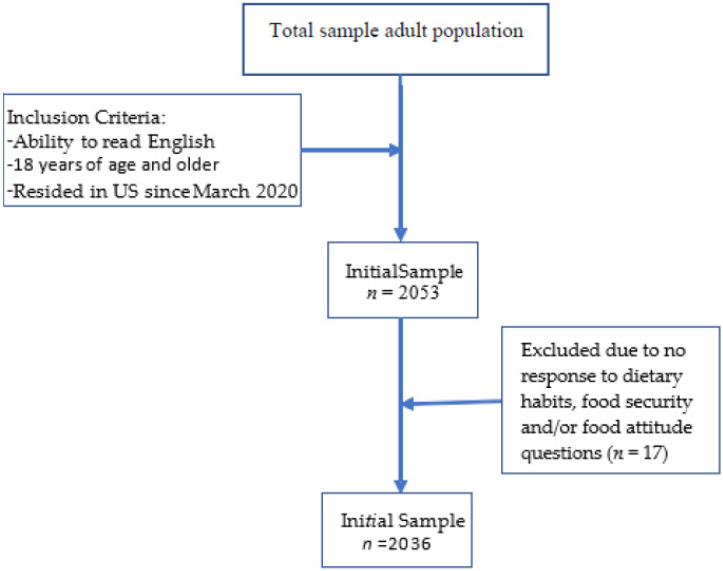
Sample collection chart.

**Table 1 nutrients-14-04636-t001:** Model for regression analysis.

Y_1_ = b_0_ + b_1 × 1_ + b_2_X_2_ + … + b_k_X_k_	
where	
Y_1_ represents	Dietary Habits
b_0_, b_1,_ and b_k_ represent	Estimate regression parameters
X_1,_ X_2,_ and X_k_ represent	k predictors (demographics, lifestyle habits, food attitudes, and food security status)

**Table 2 nutrients-14-04636-t002:** Participants’ demographics.

Variables	No. of Responses (%) ^a^
Sex	*n* = 2004
Male	414 (20.7%)
Female	1545 (77.1%)
Other	45 (2.2%)
Race/Ethnicity	*n* = 1983
African American	74 (3.7%)
Asian	59 (3.0%)
White	1696 (85.5%)
Hispanic	87 (4.4%)
Native American	12 (0.6%)
Other	55 (2.8%)
Age	*n* = 2012
18–24 years	159 (7.9%)
25–29 years	191 (9.5%)
30–49 years	675 (33.5%)
50–59 years	333 (16.6%)
60–69 years	397 (19.7%)
>70 years	257 (12.8%)
Education level	*n* = 2011
No schooling completed	1 (0.0%)
Some high school, no diploma	7 (0.3%)
High school graduate, diploma, or equivalent (GED ^b^)	62 (3.1%)
Some college credit, no degree	241 (12.0%)
Trade/technical/vocational training	78 (3.9%)
Associate degree	148 (7.4%)
Bachelor’s degree	692 (34.4%)
Master’s degree	572 (28.4%)
Professional degree	74 (3.7%)
Doctorate degree	136 (6.8%)
Current employment status	*n* = 2012
Full time	915 (45.5%)
Part-time	285 (14.2%)
Unemployed	275 (13.7%)
Other	537 (26.7%)
Marital status	*n* = 2008
Married	968 (48.2%)
Single	618 (30.8%)
Widowed	75 (3.7%)
Divorced	267 (13.3%)
Other	80 (4.0%)
People live in the household besides yourself	*n* = 2036
None	386 (19.0%)
1	808 (39.7%)
2	358 (17.6%)
3	250 (12.3%)
4	123 (6.0%)
5 or more	68 (3.3%)
Did not respond	43 (2.1%)
Currently staying at home x% of the time	*n* = 2015
Less than 25%	413 (20.5%)
50–75%	781 (38.8%)
75–95%	783 (38.9%)
Never left the house	38 (1.9%)
Residence	*n* = 2010
New England (Connecticut, Maine, Massachusetts, Rhode Island, Vermont)	80 (4.0%)
Mid-Atlantic (New Jersey, New York, Pennsylvania)	235 (11.7%)
South Atlantic (Delaware, Florida, Georgia, Maryland, North Carolina, South Carolina, Virginia, Washington DC, West Virginia)	431 (21.4%)
East North Central (Illinois, Indiana, Michigan, Ohio, Wisconsin)	382 (19.0%)
East South Central (Alabama, Kentucky, Mississippi, Tennessee)	222 (11.0%)
West North Central (Iowa, Kansas, Minnesota, Missouri, Nebraska, North Dakota, South Dakota)	174 (8.7%)
West South Central (Arkansas, Louisiana, Texas)	90 (4.5%)
Mountain (Arizona, Colorado, Idaho, Montana, Nevada, New Mexico, Utah, Wyoming)	145 (7.2%)
Pacific (Alaska, California, Hawaii, Oregon, Washington)	251 (12.5%)

Note. ^a^ Participants were not mandated to complete the demographic information of the survey. ^b^ GED = General Educational Development.

**Table 3 nutrients-14-04636-t003:** Participants’ general health characteristics and anthropometrics.

Variables	No. of Responses (%) ^a^
BMI (kg/m^2^) ^b^	*n* = 2012
<18	60 (3.0%)
18.5–24.9	794 (39.5%)
25–29.9	580 (28.8%)
>30	578 (28.7%)
Self-reported Weight change	*n* = 2035
No change	639 (31.4%)
Increased	908 (44.6%)
Decreased	488 (24.0%)
Activity	*n* = 2016
No change	626 (31.1%)
Increased	537 (26.6%)
Decreased	853 (42.3%)
Tried a diet	*n* = 2034
No	1279 (62.9%)
Yes	755 (37.1%)
Nutritional supplement intake	*n* = 2032
No	1399 (68.8%)
Yes	633 (31.2%)
Supplements currently taking	*n* = 600
Multi-vitamin	42 (7.0%)
Vitamin B complex	1 (0.2%)
Vitamin C	4 (0.7%)
Vitamin D	26 (4.3%)
Other	39 (6.5%)
Two supplements	138 (23.0%)
Three supplements	94 (15.7%)
Four or more supplements	256 (42.7%)
Medical conditions	*n* = 1403
Cancer	14 (1.0%)
Depression	234 (16.7%)
Diabetes (high blood sugar)	30 (2.1%)
Diverticulosis/Diverticulitis	8 (0.6%)
Gastric reflux	49 (3.5%)
Heart disease	87 (6.2%)
IBS/D ^c^	26 (1.9%)
Liver disease (cirrhosis, fatty liver)	2 (0.1%)
Lung disease	10 (0.7%)
Nausea/Vomiting	3 (0.2%)
Other	178 (12.7%)
2 conditions	410 (29.2%)
3 or more conditions	352 (25.1%)

Note. ^a^ Participants were not mandated to complete the survey’s health and/or anthropometrics; ^b^ BMI = Body Mass Index; ^c^ IBS/D = Irritable Bowel Syndrome/Disorder.

**Table 4 nutrients-14-04636-t004:** Frequency and counts of foods/beverages consumed since March 2020 (*n* = 2036).

Food/Beverage Items	Increased (%)	Decreased (%)	No Change/Never Consumed (%)
Milk and non-milk	411 (20.2%)	247 (12.1%)	1378 (67.7%)
Margarine or butter	212 (10.4%)	286 (14.0%)	1538 (75.5%)
Fruit	647 (31.8%)	333 (16.4%)	1056 (51.9%)
Fruit juice	229 (11.2%)	263 (12.9%)	1544 (75.8%)
Non-starchy vegetables	676 (33.2%)	248 (12.2%)	1112 (54.6%)
Vegetable or tomato juice	97 (4.8%)	131 (6.4%)	1808 (88.8%)
Eggs, chicken, or turkey	583 (28.6%)	243 (11.9%)	1210 (59.4%)
Beef, pork, or lamb	226 (11.1%)	562 (27.6%)	1248 (61.3%)
Processed meats	502 (24.7%)	313 (15.4%)	1221 (60.0%)
Fish and shellfish	424 (20.8%)	267 (13.1%)	1345 (66.1%)
Cold breakfast cereals	298 (14.6%)	395 (19.4%)	1343 (66.0%)
White bread	321 (15.8%)	289 (14.2%)	1426 (70.0%)
Dark bread	260 (12.8%)	246 (12.1%)	1530 (75.1%)
French fried potatoes	406 (19.9%)	318 (15.6%)	1312 (64.4%)
Potatoes	404 (19.8%)	254 (12.5%)	1378 (67.7%)
Starchy vegetables	388 (19.1%)	205 (10.1%)	1443 (70.9%)
White rice or pasta	306 (15%)	500 (24.6%)	1230 (60.4%)
Brown rice or whole-grain pasta	384 (18.9%)	178 (8.7%)	1474 (72.4%)
Potato chips or other salty snacks	373 (18.3%)	642 (31.5%)	1021 (50.1%)
Nuts or seeds	691 (33.9%)	181 (8.9%)	1164 (57.2%)
Peanut butter or other nut butter	512 (25.1%)	222 (10.9%)	1302 (63.9%)
Sweets	318 (15.6%)	783 (38.5%)	935 (45.9%)
Oils	313 (15.4%)	109 (5.4%)	1614 (79.3%)
Water	855 (42.0%)	145 (7.1%)	1036 (50.9%)
Coffee or Tea	182 (8.9%)	719 (35.3%)	1135 (55.7%)
Immune enhancing beverages	288 (14.1%)	41 (2.0%)	1707 (83.8%)
Beer or wine	308 (15.1%)	454 (22.3%)	1274 (62.6%)
Hard liquor	281 (13.8%)	337 (16.6%)	1418 (69.6%)
Low-calorie carbonated beverages	163 (8.0%)	296 (14.5%)	1577 (77.5%)
Carbonated beverages	241 (11.8%)	241 (11.8%)	1554 (76.3%)

**Table 5 nutrients-14-04636-t005:** Multiple linear regression of food attitudes, food security, demographics, and total dietary habit scores (*n* = 2036).

Total Dietary Habits Score	Coef.	Std. Err.	t	*p* < |t| *	95% Conf. Interval	
Food attitudes ∗ Food security	10.35	0.27	11.08	<0.001	0.73	1.04
Food attitudes score	1.11	0.09	11.84	<0.001	0.93	1.29
Food security score	0.53	0.09	5.72	<0.001	0.35	0.71
Sex: female	−1.97	0.52	−3.81	<0.001	−2.98	−0.95
Ethnicity	−0.05	0.29	−0.16	0.87	−0.62	0.52
Residence	−0.15	0.10	−1.59	0.11	−0.34	0.04
Education	−0.15	0.14	−1.05	0.29	−0.44	0.13
Employment	0.10	0.29	0.46	0.65	−0.31	0.50
Marital status	0.23	0.19	1.20	0.23	−0.15	0.61
% of time spent at home	−0.31	0.31	−0.98	0.33	−0.92	0.31
Age range	−0.16	0.20	−0.80	0.424	−0.56	0.23
Household size	−0.19	0.18	−1.06	0.291	−0.53	0.16
BMI	−0.01	0.27	−0.04	0.969	−0.53	0.51
Weight change	0.32	0.32	1.00	0.32	−0.31	0.96
Medical conditions	0.13	0.05	2.62	0.01	0.03	0.22
Tried a diet	1.53	0.49	3.10	<0.001	0.56	2.49
Nutritional supplement intake	2.55	0.50	5.10	<0.001	1.57	3.54

Food attitudes ∗ Food security is the interaction variables on dietary habits; * *p* < 0.05 is considered statistically significant.

**Table 6 nutrients-14-04636-t006:** Multiple linear regression of impact of other attributes on total dietary habits (*n* = 2036).

Attributes	Coef.	Std. Err.	t	*p* < |t| *	95% Conf. Interval	
Physical activity	1.24	0.32	3.87	<0.001	0.61	1.87
Dining at restaurants	0.17	0.35	0.47	0.63	−0.52	0.86
Preparing/cooking meals in the home	0.99	0.30	3.35	<0.001	0.41	1.57
Meal kit services	1.18	0.36	3.30	<0.001	0.48	1.88
Take-out/delivery of meals from restaurants	1.06	0.37	2.86	<0.001	0.33	1.78
Grocery shopping in the store	1.14	0.32	3.58	<0.001	0.52	1.76
Grocery shopping online	−0.14	0.24	−0.59	0.56	−0.62	0.34
Reading/studying	0.81	0.25	3.20	<0.001	0.31	1.30
Sleeping hours and quality	1.09	0.30	3.66	<0.001	0.50	1.67
Smoking (cigarettes, cigars, hookah)	2.13	0.54	3.93	<0.001	1.07	3.20
Socializing outside the home	0.06	0.46	0.14	0.89	−0.84	0.96
Using electronic devices	2.04	0.53	3.87	<0.001	1.01	3.08

* *p* < 0.05 is considered statistically significant.

## Data Availability

Data is contained within the article and can be available upon request.

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
