# Peer review of "Association between Dietary Habits, Food Attitudes, and Food Security Status of US Adults since March 2020: A Cross-Sectional Online Study"

_nutrients, 2022, doi:10.3390/nu14214636_

Round 1

Reviewer 1 Report

The manuscript entitled ‘Association between dietary habits, food attitudes, and food security status of US adults since March 2020: A cross-sectional online study ‘ presents important issue, however some corrections are needed. 

      Abstract – ‘associations using JMP SAS at a statistical significance level of p<0.05.’ – this part of sentence is not necessary in abstract.

      The purpose of the study should be presented at the end of the introduction section.

      ‘Participants responded to a 71-item survey that was used in a previous COVID-19 85 study[7].’ – please add some more detailed infoamion about the tool - more information is needed about the validity and reliability of each measure. Additionally, any limitations in reliability and validity need to be addressed in the discussion.

      Moreover, if authors modified questionnaire (lines 89-90) ‘Compared to the survey conducted during confinement in 2020, additional questions were added to the questionnaire…” – the validation must be conducted once again,

      The share of female (77.1%) is high – how it could influence the obtained results?

      Figure 2 – it is difficult to follow. Maybe table is more suitable.

      “Second, as a result of participants not being required to answer every survey item and self-reporting information, it could have led to self-reporting bias.” Please add also self-reported height and weight

Author Response

The manuscript entitled ‘Association between dietary habits, food attitudes, and food security
status of US adults since March 2020: A cross-sectional online study ‘ presents important issue,
however some corrections are needed.

We appreciate this positive comment. We have made attempts to address each of your
suggestions as indicated below. We cannot thank you enough for taking the time to review this
manuscript.

Abstract ‘associations using JMP SAS at a statistical significance level of p<0.05.’ this
part of sentence is not necessary in abstract.

We have subsequently removed this part.

The purpose of the study should be presented at the end of the introduction section.

We have restructured the introduction to include the purpose statement near the end of the
introduction section.

‘Participants responded to a 71-item survey that was used in a previous COVID-19 85
study[7].’ please add some more detailed infoamion about the tool - more information is
needed about the validity and reliability of each measure. Additionally, any limitations in
reliability and validity need to be addressed in the discussion.

Thank you for raising this point. We have revised aspects of this method to expand on these
instruments without causing significant overlap with a prior publication of ours. Line 88 of the
methods section indicates about the adapted validated instruments. Per the comments we have
detailed below, since all the instruments were considered reliable with Cronbach alphas >0.6
and valid with correlation coefficients for each item >0.2, we did not explicitly address
limitations with these in the discussion section.

Moreover, if authors modified questionnaire (lines 89-90) ‘Compared to the survey
conducted during confinement in 2020, additional questions were added to the
questionnaire...” the validation must be conducted once again,

We reduced the first paragraph of the survey information so we could provide further
information about the validation/reliability of these instruments within each survey section. For
example, in lifestyle habits, we performed a reliability analysis of this modified instrument as
well as validation in lines 104-106. We subsequently performed the same analyses for dietary
habits (lines 122-124), food attitudes (lines 134-136) and food security lines 143-144).

The share of female (77.1%) is high how it could influence the obtained results?

Thank you for raising this very important point. In the discussion, we elaborated more on this
idea lines 232-245.

Figure 2 it is difficult to follow. Maybe table is more suitable.

We have removed that figure and created a table, 4.
“Second, as a result of participants not being required to answer every survey item and
self-reporting information, it could have led to self-reporting bias.” Please add also self-
reported height and weight

Thank you for pointing this out, we included the information about height and weight (line 347)
in the limitation area of the discussion.

Reviewer 2 Report

Thank you for the opportunity to review this manuscript, examining food habits and food security in the United States as affected by the COVID-19 pandemic. This topic is of high importance and the study addresses an important literature gap, with there currently being little available evidence as to the effects of the pandemic on dietary habits and food security.

I have outlined a few comments to help clarify several aspects of the manuscript:

Abstract/overall:

·        Overall an interesting study. Studies examining the dietary effects of the pandemic are still uncommon so it is good to see a study with robust methods

Introduction:

·        Line 36: citation needed for the statistic of 26 million jobs lost

·        Line 36-39: this could be clarified. A reduction in unemployment from 2021 to 2022 is a good thing, correct? The way the sentence reads is as if this is part of the severe impacts on the labour market

·        Line 49: sentence about home cooking should not begin with “furthermore” after discussing increasing consumption of foods high in saturated fat, sugar, etc. Home cooking has been shown to be healthier than purchasing food, so doesn’t seem logical to raise concerns about increased fat and sugar in this context

·        Line 61: concerns about sleep needs a citation

Methods:

·        Line 89: citation needed for “validated instruments” used to develop the survey

·        Methods for assessing dietary habits, food attitudes, and food security described well

Discussion:

·        Line 228: do you have any theories as to why female sex was negatively associated with dietary habit scores?

·        Line 282: healthiness of meal kits, I suggest adding a citation to this study which discusses the nutritional profile of meal kits - Gibson AA, Partridge SR. Nutritional Qualities of Commercial Meal Kit Subscription Services in Australia. Nutrients. 2019; 11(11):2679. https://doi.org/10.3390/nu11112679

·        Line 288: using terminology “low food security scores” is confusing. On your assessment tool the score was low, but food security for this population was actually high. Rephrase to clarify this.

·        Line 292: citation needed for increase in store food prices between 2021 and 2022

·        Paragraph 288 – 301: Food security was high for your population group but this group was not representative of the US population (higher education, mostly lived with 1 other person only). A comment here about how this may not reflect the experiences of the wider population during the pandemic.

·        Line 294: prices have increased the most for food items that were frequently consumed by participants in this study – perhaps because study sample was mostly full time working highly educated single people, not as affected by food prices as other groups

·        Line 303: most participants did not shop online – is this the case in other literature?

·        Line 306: citation for “adults prefer the atmosphere and experience of shopping in physical stores”

·        Line 307: citation for “The ability to personally select food is the primary driver of the in-store advantage”

·        Line 307: suggest different word selection instead of “advantage”

·        Line 309: sentence does not flow from previous point, psychological effects not discussed in this paragraph

·        Line 324: add self reporting inaccuracies of BMI

Conclusion:

·        Line 338: further studies could also investigate whether these habits have been maintained

Author Response

Thank you for the opportunity to review this manuscript, examining food habits and food
security in the United States as affected by the COVID-19 pandemic. This topic is of high
importance and the study addresses an important literature gap, with there currently being
little available evidence as to the effects of the pandemic on dietary habits and food security.

We appreciate that you took the opportunity to review our manuscript and provide a review
that only helps improve our manuscript. We have addressed your comments to the best of our
abilities.

I have outlined a few comments to help clarify several aspects of the manuscript:

Abstract/overall:

· Overall an interesting study. Studies examining the dietary effects of the pandemic are still
uncommon so it is good to see a study with robust methods

Thank you for these thoughts.

Introduction:

· Line 36: citation needed for the statistic of 26 million jobs lost

We have included the appropriate citation in that line.

· Line 36-39: this could be clarified. A reduction in unemployment from 2021 to 2022 is a
good thing, correct? The way the sentence reads is as if this is part of the severe impacts on the
labour market

Thank you for raising this point, we have revised this statement so it is clearer our intention.

· Line 49: sentence about home cooking should not begin with “furthermore” after
discussing increasing consumption of foods high in saturated fat, sugar, etc. Home cooking has
been shown to be healthier than purchasing food, so doesn’t seem logical to raise concerns
about increased fat and sugar in this context

Thank you for identifying this grammatical error. We have subsequently modified our
transition to align better with a different concept that we are introducing.

· Line 61: concerns about sleep needs a citation

We have added the appropriate reference

Methods:

· Line 89: citation needed for “validated instruments” used to develop the survey

We have subsequently added these references.

· Methods for assessing dietary habits, food attitudes, and food security described well

Discussion:
· Line 228: do you have any theories as to why female sex was negatively associated with
dietary habit scores?

Thank you for raising this important point. We have provided some thoughts, which can be
found in lines 232-245.

· Line 282: healthiness of meal kits, I suggest adding a citation to this study which discusses
the nutritional profile of meal kits - Gibson AA, Partridge SR. Nutritional Qualities of
Commercial Meal Kit Subscription Services in Australia. Nutrients. 2019; 11(11):2679.
https://doi.org/10.3390/nu11112679

Thank you for providing this to us. We also expanded on bit more on this concept in lines 297-
306.

· Line 288: using terminology “low food security scores” is confusing. On your assessment
tool the score was low, but food security for this population was actually high. Rephrase to
clarify this.

Thank you for pointing this out. We have modified our discussion to ensure that we are
acknowledging our participants were considered food secure as opposed to low food insecurity.

· Line 292: citation needed for increase in store food prices between 2021 and 2022

We have provided this information.

· Paragraph 288 301: Food security was high for your population group but this group was
not representative of the US population (higher education, mostly lived with 1 other person
only). A comment here about how this may not reflect the experiences of the wider population
during the pandemic.

Great point. We have modified this section to include this very important finding of our
population in now what is lines 318-324.

· Line 294: prices have increased the most for food items that were frequently consumed by
participants in this study perhaps because study sample was mostly full time working highly
educated single people, not as affected by food prices as other groups

We slightly modified this statement to coincide with now lines 318-324 to minimize repetition of
information discussed.

· Line 303: most participants did not shop online is this the case in other literature?

We revised the information we included initially about the study performed in China about on-
line vs in-person grocery shopping since the pandemic to convey more about what we intended
for this information.

· Line 306: citation for “adults prefer the atmosphere and experience of shopping in physical
stores”

completed

· Line 307: citation for “The ability to personally select food is the primary driver of the in-
store advantage”

completed

· Line 307: suggest different word selection instead of “advantage”

revised

· Line 309: sentence does not flow from previous point, psychological effects not discussed
in this paragraph

We modified and removed the information about psychological effects.

· Line 324: add self reporting inaccuracies of BMI

Thank you for mentioning this information. We have added this in line 348.

Conclusion:

· Line 338: further studies could also investigate whether these habits have been maintained

Well-taken point, which we have included in line 363.

Reviewer 3 Report

The pape is well written. The research question is very clear, as is the investigation method. The results obtained suggest that encouraging healthy habits (dietary, physical activity etc.) should always guide public policies, regardless of specific events such as COVID-19. The differences obtained when compared to other studies are probably due to the eating habits of each region. In the text, the comments in yellow were to accompany may assessment.

Author Response

Thank you for your generous comments. We greatly appreciate the time you took to review our manuscript.